# Effects of Temperature on the Timeliness of eDNA/eRNA: A Case Study of *Fenneropenaeus chinensis*

**Tangyi Qian** [1,2], **Xiujuan Shan** [1,2,3,*], **Weiji Wang** [1,2,3] **and Xianshi Jin** [1,2,3]

1    Key Laboratory of Sustainable Development of Marine Fisheries, Ministry of Agriculture and Rural Affairs, Shandong Provincial Key Laboratory of Fishery Resources and Ecological Environment, Yellow Sea Fisheries Research Institute, Chinese Academy of Fishery Sciences, Qingdao 266071, China; qiantangyi19@126.com (T.Q.); wangwj@ysfri.ac.cn (W.W.); jin@ysfri.ac.cn (X.J.)

2    Laboratory for Marine Fisheries Science and Food Production Processes, Pilot National Laboratory for Marine Science and Technology (Qingdao), Qingdao 266237, China

3    National Field Observation and Research Center for Fisheries Resources in Changdao Coastal Waters, Changdao 265800, China

*    Correspondence: shanxj@ysfri.ac.cn

**Abstract:** Environmental DNA (eDNA) technology has been successfully applied to detect organisms in various aquatic ecosystems. However, eDNA has been proven to exist for a long time in environmental samples. The timeliness of eDNA detection results largely depends on the rate of molecular degradation. Environmental RNA (eRNA) is considered an excellent complementary tool because most researchers believe that RNA degrades faster than DNA in vitro, while, to the best of our knowledge, the number of published articles related to eRNA is very limited. To address an important knowledge gap, this study focused on the response mechanism of eRNA degradation to water temperature change as compared with eDNA. Changes in the concentration of eDNA and eRNA of the mitochondrial cytochrome c oxidase subunit 1 (COI) gene from *Fenneropenaeus chinensis* were detected at four temperatures (10, 15, 20 and 25 °C). The results showed that the degradation rate of eDNA increased with an increase in temperature. The degradation rate constants ranged from 0.011 to 0.486 h$^{-1}$ and the degradation time ranged from 8 to 383 h for eDNA. The degradation rate of eRNA changed slightly with an increase in temperature. The degradation rate constants ranged from 0.190 to 0.379 h$^{-1}$ and the degradation time ranged from 11 to 22 h for eRNA. eRNA showed better stability under temperature change and maintained a faster degradation rate at low temperatures. These results provide answers to the questions of whether eRNA and eDNA degradation rates are fast or slow. Furthermore, this study may suggest the potential superiority of eRNA over eDNA and promote further study of eRNA in future research.

**Keywords:** biomonitoring; environmental DNA; environmental RNA; degradation rate; water temperature

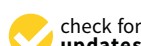



## 1. Introduction

Biological monitoring plays an important role in ecosystem management and conservation [1]. Environmental DNA (eDNA) technologies, as a low-cost, non-invasive survey method, have developed rapidly in the past decade, becoming supplementary to traditional survey methods. Additionally, they have been widely used in the biological monitoring of various aquatic organisms [2–5]. The amount of eDNA-related research has increased rapidly, and many research institutions around the world are integrating eDNA technology into their investigative and monitoring practices [6–9]. Different from traditional investigative methods, where the existence or abundance of target organisms is determined by capturing or recording individuals [10], eDNA is used to identify individuals by detecting DNA sequences released into the environment and does not require morphological expertise and experience [11,12]. In addition, eDNA is ubiquitous in the natural environment, and can come from animal feces, urine, mucus and shed cells [2]. Therefore, the false

negative results caused by undetected rare taxa which can occur when using traditional investigation methods are avoided [10,13].

Although eDNA technologies are widely used in biological resource assessments and biodiversity surveys, the results are often ambiguous [14,15]. eDNA remains in the natural environment for a long time after its release [16], and it can be transferred long distances along the water column, making it difficult to trace the source of the studied species [17]. Additionally, the ability to effectively distinguish between living (metabolic activity or dormancy) and dead organisms is often particularly important when protecting rare wild species and controlling biological invasion [18,19]. Finding a molecule that is similar to eDNA and has a faster turnover rate could be an effective solution to this problem. Some researchers have observed that in vitro RNA is a much less stable molecule than DNA and degrades rapidly when it is shed from an organism [20,21]. Based on this feature, it is possible to use environmental RNA (eRNA) in biological monitoring. However, published articles contain different opinions on the timeliness of eRNA. For example, Wood et al. [22] found no significant difference in the degradation rates between eDNA and eRNA from *Sabella spallanzanii* and *Styela clava* and it was believed that the associated eRNA remained in the natural environment longer than expected. In contrast, Marshall et al. [23] found that the eRNA degradation rate in mussels was higher than that of eDNA, and that the eRNA was more time-sensitive. However, these studies did not take into account the possible influence of environmental factors. This is important because the degradation rates of eDNA/eRNA may be different under different environmental conditions.

eDNA/eRNA is mainly susceptible to acidic conditions or hydrolysis by enzymes. Exogenous nucleases produced by microbial activity can decompose eDNA/eRNA in the water into short fragments until they cannot be detected, and changes in environmental factors directly affect the hydrolysis reactions of related enzymes [24]. For example, temperature, UV-B and pH affect the degradation rate of eDNA, among which temperature has the most significant impact [25]. High temperatures (>50 °C) directly cause DNA/RNA denaturation and degradation, which is why it is difficult for organisms to survive at high temperatures. However, it has been confirmed that some organisms, referred to as thermophilic organisms, can survive at high temperatures [26]. In most cases, temperature-related eDNA/eRNA degradation may be caused by indirect effects. For example, moderately high temperatures will stimulate biological metabolism and exonuclease activity [27–31]. Therefore, eDNA/eRNA degradation rates may be different at different temperatures.

Chinese prawns (*Fenneropenaeus chinensis*) (Crustacea, Decapoda), large benthic arthropods, are an important fishery resource in the Yellow Sea and Bohai Sea of China. Due to the impact of human activities, the wild resources of *F. chinensis* have declined seriously [32]. By the 1990s, *F. chinensis* almost completely relied on artificial proliferation and release for its maintenance as a resource [33]. Hence, the key to formulating appropriate release strategies is to accurately evaluate the proliferation effect of *F. chinensis*. It is necessary to understand and master the real-time population and specific distribution of *F. chinensis*. However, the traditional fishery resources survey methods cannot accurately reflect the population dynamics and temporal and spatial distribution of *F. chinensis*. It has been difficult to evaluate its resources using traditional investigation methods [34]. Therefore, it is urgent to identify a more suitable method for the study of *F. chinensis*.

In this study, *F. chinensis* was selected as the research object. The degradation rates and residual time of the eDNA and eRNA molecules of *F. chinensis* were analyzed at four temperatures (10, 15, 20 and 25 °C) which cover the range that *F. chinensis* can adapt to in the wild [35]. Finally, the difference between the temperature sensitivity of eDNA and eRNA is discussed. We use the term eDNA and eRNA to refer to nucleic acid fragments or tissue fragments found in the environment, not individual protozoans, algae or bacteria. The following topics are discussed: the degradation rates of eDNA and eRNA at different temperatures and the differences in timeliness and temperature sensitivity between eDNA and eRNA. In this study, temperature was included as a variable for the first time when

researching the degradation process of eRNA. Accordingly, this research addresses some problems existing in current eRNA research, promotes the progress of eRNA methodology and improves the reliability of environmental nucleic acid analysis.

## 2. Materials and Methods

### 2.1. Experimental Setup

The twenty *F. chinensis* (average length: 181.7 $\pm$ 6.74 mm, average wet weight: 55.03 $\pm$ 5.29 g) used in the laboratory experiment were caught by trawling the sea study area (Qingdao, Shandong Province, China; 10 December 2020). *F. chinensis* were transported back to the laboratory in PE bags filled with seawater and carefully rinsed with prepared artificial seawater (30‰ salinity) to remove residues from their surface. Without feeding, the *F. chinensis* were moved into a tank containing seawater for 24 h to acclimate to the environment. The water temperature was maintained at 15 $\pm$ 1 °C and continuously aerated. According to our previous experience, the activity of *F. chinensis* at this temperature is moderate. The experimental setup was as follows: at 10:00 a.m. on 11 December 2020, twenty *F. chinensis* were carefully moved into a 65 L tank (length, width and height: 58.5 $\times$ 44 $\times$ 34.5 cm). The tank was filled with 50 L of natural seawater purified by sun exposure and filtration with 0.45 µm glass fiber filters, maintained at 15 $\pm$ 1 °C and continuously aerated using an aerator pipe. The *F. chinensis* were kept in the tank without feeding for 24 h before being removed. At 10:00 a.m. the next day, the *F. chinensis* were removed from the tank and the seawater was mixed using an aerator pipe and then divided into four experimental groups. Each seawater group (10 L) was placed into a 15 L tank (length, width and height: 35 $\times$ 24 $\times$ 20 cm). The temperatures of the seawater, each representing one of the four experimental groups, were maintained at 10, 15, 20 and 25 °C. This range was divided into four levels for better analysis of the changing trend. The groups were continuously aerated, and the other conditions remained completely the same. The four experimental groups were covered with PE film to avoid seawater evaporation due to the high temperatures. The removed *F. chinensis* were fed with short-necked clams (*Ruditapes philippinarum*) to maintain physiological activity and moved into another 65 L water tank to continue the experiment. In addition, two replicate experiments were carried out for comparison. All experimental equipment was bleached with a 0.1% potassium hypermanganate solution, followed by multiple rinses with ddH$_2$O.

### 2.2. Sample Collection and Processing

In order to verify that there was no contamination before the experiment, 9 L water samples were collected from the 65 L tanks prior to the addition of *F. chinensis*. Time 0 was defined as the time immediately after the *F. chinensis* were removed from the 65 L tank. Water samples were taken from each tank at the following time points: 0, 4, 8, 12, 24, 72, 120 (5th day), 168 (7th day), 336 (14th day) and 504 (21st day) [22]. Aerator pipes were used to gently swirl the water in the tanks before sampling to confirm seawater mixing. Water samples (3 L) were collected using sterile 1 L beakers.

Water samples were filtered immediately after collection using polycarbonate filters (3 µm, 47 mm dia.; Whatman, Shanghai, China). Between each sample, the experimental equipment was wiped with 75% alcohol and subjected to high-temperature disinfection, followed by rinsing with ddH$_2$O and drying. After water filtration, the filter paper was transferred into sterile 1.5 mL tubes (Axygen BioScience, Corning, NY, USA) with 1 mL of RNA stabilizer (Vazyme Biotech, Nanjing, China) and then stored at −20 °C until RNA and DNA extraction. After sampling was completed, all samples were extracted uniformly (Figure 1). During sample collection, temperature-control equipment was used to continuously monitor the seawater. Other physicochemical parameters (salinity, light and dissolved oxygen) remained the same.

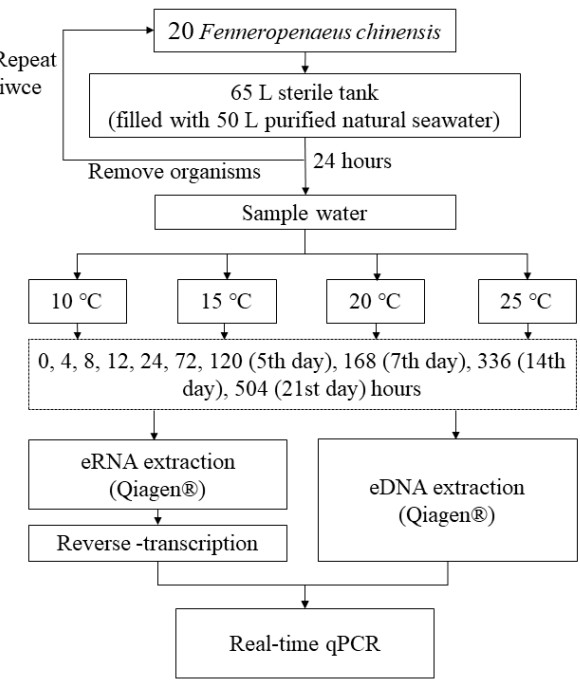

**Figure 1.** Flow chart describing the four different treatment protocols applied to *Fenneropenaeus chinensis*-associated eRNA and eDNA purification and detection. After the *F. chinensis* were removed, the water samples were treated at different temperatures (10, 15, 20 and 25 °C) and sampled at 0, 4, 8, 12, 24, 72, 120, 168, 336 and 504 h for filtration and subsequent operations.

### 2.3. eDNA and eRNA Extraction and cDNA Synthesis

After the filter paper was broken and homogenized, eDNA and eRNA were then co-extracted using the QIAGEN$^{TM}$ AllPrep DNA/RNA Mini Kit (QIAGEN, Hilden, Germany), following the manufacturer's protocol. Total DNA and RNA was eluted in 100 µL of DNase/RNase-free water. Extracted DNA and RNA were tested for quality and purity using an ultra-trace spectrophotometer (LifeReal, Hangzhou, China), then the DNA was stored at −20 °C until subsequent operations, while the RNA was immediately subjected to cDNA synthesis and the remaining RNA was stored at −80 °C. Each set of extractions included a negative extraction control.

Reverse transcription of RNA to cDNA was carried out using HiScript$^{®}$III RT Super-Mix (Vazyme Biotech, Nanjing, China), which includes a random primer/Oligo (dT)20 VN primer mix. cDNA was stored at −20 °C until subsequent analysis. The amount of cDNA product represents the amount of eRNA. To confirm the absence of DNA in the RNA eluate when a cDNA sample yielded a positive result, qPCR was performed on the RNA sample after DNase treatment to ensure that the positive result for RNA was not due to DNA contamination.

### 2.4. Quantitative Polymerase Chain Reaction

qPCRs were run using a 2×TaqMan Fast qPCR Master Mix (Low Rox) real-time PCR kit (BBI Life sciences, Shanghai, China) on an Applied Biosystems RT-qPCR System (model 7500, Thermo Fisher Scientific, Waltham, MA, USA). Quantitative Analysis of Mitochondrial Cytochrome I (COI) Gene concentrations in DNA and cDNA samples using *F. chinensis*-specific primers and probes (Table 1) was performed according to a method previously described by Miao Li et al. The reaction volume was 20 µL and covered 10 µL 2 × TaqMan Fast qPCR Master Mix, 0.4 µL forward and 0.4 µL reverse primers (10 µmol/L), 0.4 µL probe (10 µmol/L), 6.8 µL PCR-grade water and 2 µL of template. The reaction began with pre-denaturation at 94 °C for 3 min, followed by 40 cycles of denaturation at 94 °C for 5 s and annealing at 60 °C for 35 s. Each plate of samples contained three wells as negative controls. The absolute quantitative method was used to analyze the experimental data.

Plasmid DNA was diluted from $10^8$ to $10^1$ copies/µL and constructed using quantitative standard curves of the targeted genes [32].

**Table 1.** Primer pairs (DF-Forward, DR-Reverse) and the probe used for qPCR analysis of environmental DNA and RNA.

| Gene | Primer | Primer Sequence (5′–3′) | Tm (°C) | Length (bp) |
|---|---|---|---|---|
| COI | DF | AGGGGTAGGAACAGGATGAAC | 57.7 | 106 |
| COI | DR | GACACCAGCTAGATGCAGCG | 59.1 | 106 |
| | Probe | 5′6-FAM-TCAGCTAGAATTGCTCATGCCGGAGCTTCAGT-3′ BHQ1 | 66.2 | 106 |

To avoid cross-contamination, each stage of the molecular analysis (DNA/RNA extractions, RNA reverse transcription, qPCR setup, template addition and qPCR analysis) was carried out in a separate sterile laboratory with a sequential work flow. Each room was equipped with a UV sterilization mechanism that was turned on for at least 15 min before and after each usage. The qPCR setup and template addition were completed in HEPA-filtered laminar flow cabinets. Throughout the study, aerosol barrier pipette tips (Axygen BioScience, Corning, NY, USA) were applied.

*2.5. Statistical Analyses*

All statistical analyses were conducted using R software (version 4.0.4) with the integrated development environment R Studio. The raw data were fitted with an exponential degradation model $N(t) = N_0 e^{-\lambda t}$ to obtain degradation rate constants. $N(t)$ is the concentration of *F. chinensis* eDNA/eRNA at time $t$, and $N_0$ is the concentration of *F. chinensis* eDNA/eRNA at time 0. The 'easynls' package developed in R software was used to fit degradation rate models [36]. Using the predicted degradation rate constant ($\lambda$), the time required for *F. chinensis* eDNA/eRNA to decay to undetectable levels was calculated, as described in Strickler et al. [25]. The Wilcoxon rank-sum test of paired samples was carried out to determine whether there were significant differences in degradation rate constants between eDNA and eRNA.

**3. Results**

*3.1. eDNA/RNA Target Gene Detection*

Using 2% agarose gel electrophoresis, we confirmed that the primer pairs successfully amplified the 106 bp target fragment of the COI gene from *F. chinensis*. The electrophoresis bands were single and visible (Figure 2), which is consistent with the expected results, indicating that the selected primers were of good specificity. The double bands and fade of cDNA during the later stage were related to the low concentration of eRNA after degradation.

During the real-time qPCR amplification process, the system drew the standard curve of the *F. chinensis* COI gene according to the change in fluorescence value (Figure 3) and automatically generated the correlation coefficient of the curve, where $R^2$ was 0.996 and the regression equation was $y = -3.2 x + 40.35$. The results show a clear linear correlation over the range of diluted plasmid standards. It is proved that rt-qPCR can accurately feedback the amplification of template DNA/cDNA.

*3.2. eDNA/eRNA Release and the Initial Concentration*

Following DNase treatment, qPCR tests were used to check for DNA contamination in RNA samples. All samples were negative. The eDNA and eRNA from *F. chinensis* were detected immediately after the removal of *F. chinensis* ($N_0$). Kruskal–Wallis tests showed that there was no significant difference between the initial concentrations of eDNA and eRNA ($p = 0.194$), which indicates that the amount of released eRNA was basically the same as that of the eDNA within the first 24 h (Figure 4, Table 2).

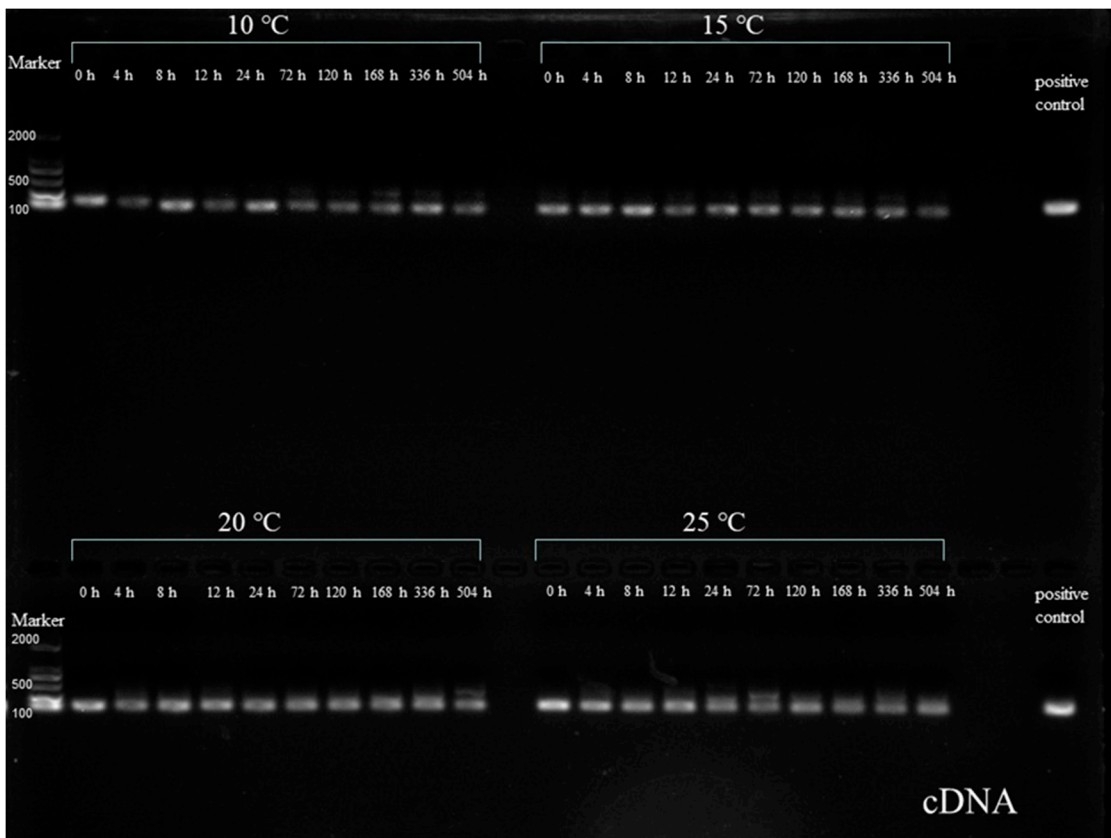

**Figure 2.** Agarose (2%) gel electrophoresis image of cDNA from reverse-transcribed eRNA from different temperature treatments (10, 15, 20 and 25 °C).

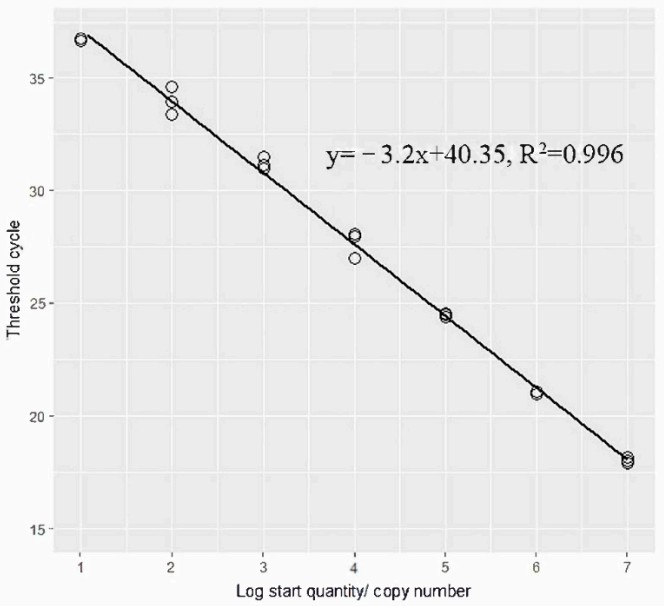

**Figure 3.** The standard qPCR curve of the *Fenneropenaeus chinensis* COI gene. The *x*-axis represents the logarithm of the plasmid DNA/cDNA concentration; the *y*-axis indicates the threshold cycle value of the qPCR.

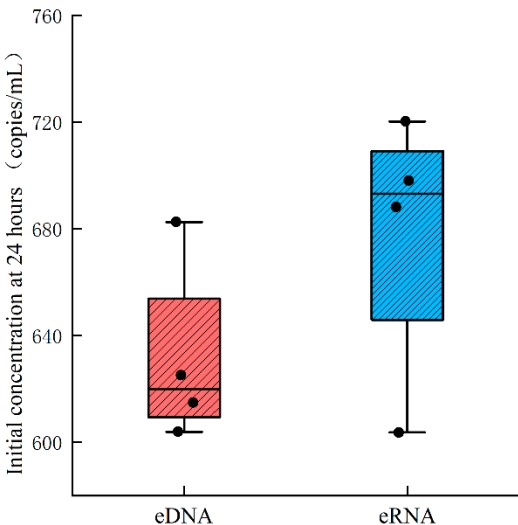

**Figure 4.** Initial concentration (copies/mL; $N_0$) of *Fenneropenaeus chinensis* eDNA and eRNA.

**Table 2.** Average concentration of *Fenneropenaeus chinensis* eDNA and eRNA at four temperatures (10, 15, 20 and 25 °C).

|  | Time (h) | 10 °C (copies/mL) | 15 °C (copies/mL) | 20 °C (copies/mL) | 25 °C (copies/mL) |
|---|---|---|---|---|---|
| eDNA | 0 | 682.56 ± 43.15 | 625.15 ± 17.06 | 603.95 ± 117.17 | 614.87 ± 78.88 |
|  | 4 | 696.78 ± 54.77 | 523.17 ± 40.53 | 157.05 ± 25.23 | 80.51 ± 11.55 |
|  | 8 | 627.12 ± 34.96 | 504.80 ± 75.79 | 105.58 ± 20.27 | 33.33 ± 31.34 |
|  | 12 | 613.97 ± 71.83 | 478.02 ± 21.10 | 63.93 ± 12.48 | 24.99 ± 1.71 |
|  | 24 | 379.07 ± 12.26 | 146.74 ± 6.44 | 33.03 ± 3.03 | 20.71 ± 3.46 |
|  | 72 | 354.70 ± 23.17 | 47.71 ± 7.69 | 5.52 ± 0.62 | 5.28 ± 1.07 |
|  | 120 | 230.43 ± 40.11 | 29.19 ± 3.37 | 3.65 ± 0.42 | 5.39 ± 1.14 |
|  | 168 | 27.13 ± 7.73 | 21.43 ± 10.42 | 5.96 ± 1.75 | 7.39 ± 2.18 |
|  | 336 | 12.17 ± 3.58 | 11.49 ± 2.36 | 8.44 ± 1.44 | 7.84 ± 2.72 |
|  | 504 | 11.22 ± 4.04 | 10.02 ± 0.90 | 7.20 ± 0.45 | 6.44 ± 0.76 |
| eRNA | 0 | 688.2 ± 27.18 | 698.07 ± 74.92 | 720.26 ± 13.13 | 603.66 ± 72.64 |
|  | 4 | 386.8 ± 23.28 | 361.19 ± 12.70 | 223.71 ± 86.35 | 135.50 ± 11.92 |
|  | 8 | 107.87 ± 72.90 | 130.36 ± 5.56 | 56.14 ± 3.86 | 22.52 ± 4.67 |
|  | 12 | 46.16 ± 2.84 | 40.96 ± 23.46 | 14.52 ± 2.21 | 7.95 ± 1.71 |
|  | 24 | 19.66 ± 3.00 | 35.29 ± 12.06 | 20.81 ± 2.01 | 8.46 ± 4.96 |
|  | 72 | 23.74 ± 2.37 | 32.66 ± 4.36 | 6.86 ± 2.28 | 8.27 ± 2.18 |
|  | 120 | 21.28 ± 4.77 | 18.27 ± 3.13 | 5.14 ± 1.83 | 8.93 ± 1.35 |
|  | 168 | 3.42 ± 1.01 | 4.82 ± 0.51 | 3.40 ± 0.24 | 3.44 ± 0.34 |
|  | 336 | 3.77 ± 1.49 | 3.33 ± 1.24 | 3.05 ± 0.48 | 3.65 ± 0.27 |
|  | 504 | 4.11 ± 0.85 | 3.68 ± 2.65 | 1.61 ± 0.42 | 3.21 ± 0.10 |

### 3.3. eDNA/eRNA Degradation Rates

The eDNA and eRNA of the four experimental groups exhibited exponential degradation (Figures 5 and 6). The degradation rate constants for eDNA varied from 0.011 to 0.486 h$^{-1}$ and from 0.190 to 0.379 h$^{-1}$ for eRNA (Table 3). Kruskal–Wallis tests showed that there was a significant difference between the degradation concentration of eDNA and eRNA at 10 °C ($p$ = 0.016 *). At higher temperatures (20 and 25 °C), there was no significant difference between the concentrations of eDNA and eRNA. This indicates that at a low temperature the degradation time of eRNA was shorter than that of eDNA, while at high temperatures the eDNA and eRNA degradation times were basically the same ($p_{15°C}$ = 0.079, $p_{20°C}$ = 0.709, $p_{25°C}$ = 0.986).

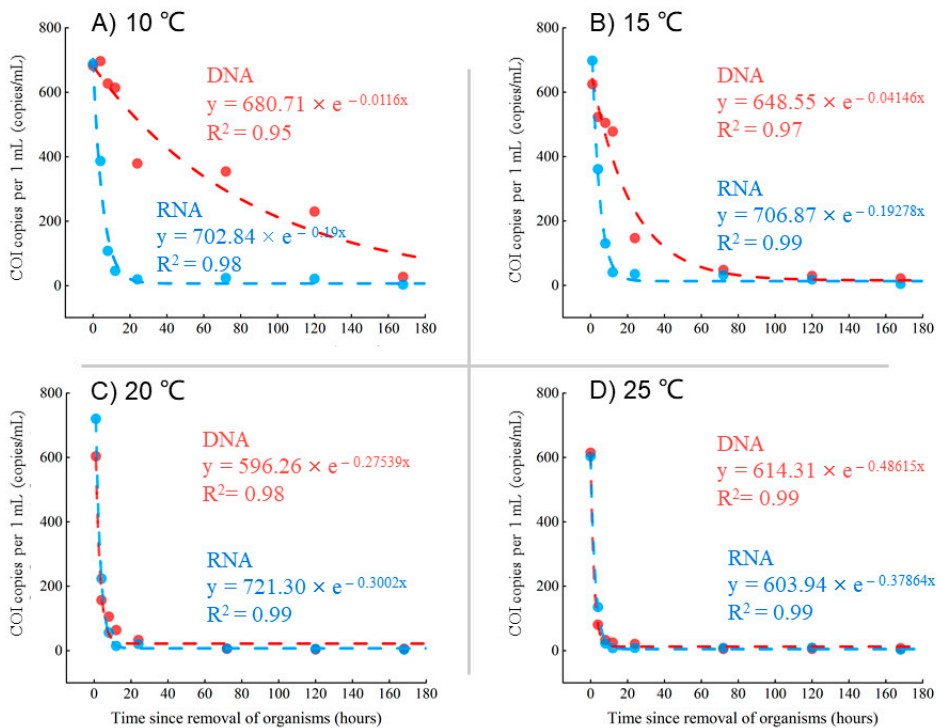

**Figure 5.** Time-dependent alterations in average eDNA and eRNA concentrations at: (**A**) 10 °C; (**B**) 15 °C; (**C**) 20 °C; and (**D**) 25 °C. After applying the degradation model N(t) = $N_0 e^{-\lambda t}$ to all raw data, the equations showed the rate of exponential degradation. $R^2$ values reflect the closeness of fit of raw data to the fitted degradation model.

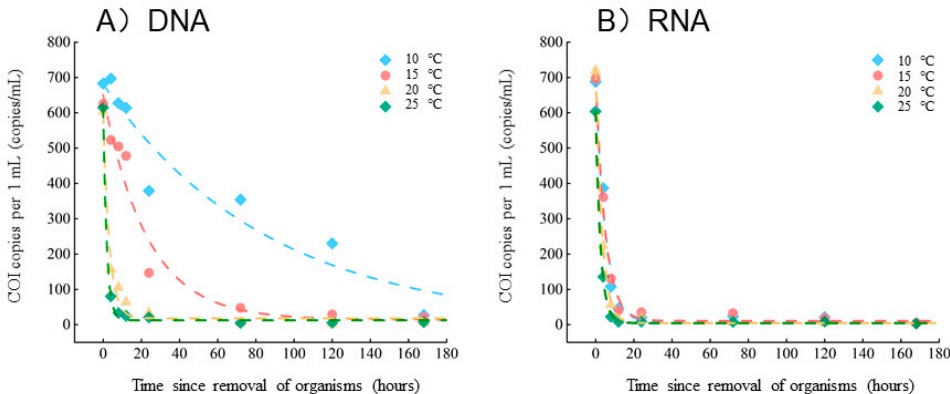

**Figure 6.** Time-dependent changes at four temperatures (10, 15, 20 and 25 °C) for: (**A**) DNA and (**B**) RNA.

**Table 3.** Degradation rate constants and estimated hours until *Fenneropenaeus chinensis* eDNA/eRNA degradation at four temperatures (10, 15, 20 and 25 °C).

| | Treatment (°C) | $N_0$ (COI Copies per mL) | Average Model-Derived Degradation Rate Constant ($\lambda$) | Predicted Time Until eDNA/eRNA Degrades below Detectable Limits |
|---|---|---|---|---|
| eDNA | 10 | $682.56 \pm 43.15$ | 0.011 | 383 |
| | 15 | $625.15 \pm 17.06$ | 0.041 | 101 |
| | 20 | $603.95 \pm 117.17$ | 0.275 | 15 |
| | 25 | $614.87 \pm 78.88$ | 0.486 | 8 |
| eRNA | 10 | $688.20 \pm 27.18$ | 0.190 | 22 |
| | 15 | $698.07 \pm 74.92$ | 0.192 | 22 |
| | 20 | $720.26 \pm 13.13$ | 0.300 | 14 |
| | 25 | $603.66 \pm 72.64$ | 0.379 | 11 |

* These values were calculated by fitting the raw data to the exponential degradation rate model $N(t) = N_0\, e^{-\lambda t}$. In this study, the detection limit for qPCR was 10 copies/mL.

## 4. Discussion

Determining the timeframe that eDNA/eRNA may be identified after the target species has departed or been removed from waters of interest, thereby inferring the time period when the target organism may have existed, facilitates the application of eDNA/eRNA technology in biomass monitoring or stock assessment [21,25]. Without considering other factors (except temperature), regression analysis showed that the degradation of eDNA and eRNA followed an exponential declining pattern at the four temperatures (10, 15, 20 and 25 °C) after the removal of *F. chinensis*. These results are consistent with what has been observed in previous studies, both for eDNA [3,6,14,16,25,37].

Although some previously published research has compared the degradation rates of eDNA and eRNA [22,23], these studies have not considered the influence of temperature on the results. Our study shows that higher water temperatures (15, 20 and 25 °C) accelerated eDNA degradation (Table 2, Figure 6A). After the removal of *F. chinensis*, eDNA degraded to below the detection limit after 383 h at 10 °C, 101 h at 15 °C, 15 h at 20 °C and 8 h at 25 °C, with the eDNA degradation rates being 0.011, 0.041, 0.275 and 0.486, respectively (Table 3). These results support previous studies that showed water temperature-dependent degradation of eDNA [9,25,38]. However, the effect of temperature on eRNA degradation was much smaller than expected (Table 2, Figure 6B). eRNA degraded to below the detection limit after 22 h at 10 °C, 22 h at 15 °C, 14 h at 20 °C and 11 h at 25 °C, with the eRNA degradation rates being 0.190, 0.192, 0.300 and 0.379, respectively (Table 3). It is apparent that eDNA is more susceptible to temperature than eRNA. The eRNA and eDNA degradation rates were different at different temperatures, without considering the external conditions; hence, it is not appropriate to compare the degradation rates of eRNA and eDNA. Our study found that, even at a low temperature, the eRNA degradation time was maintained within the first 24 h. Therefore, when eRNA is detected in natural water, we can speculate that living target organisms may have passed through within 24 h. We can also track the target organisms based on the water flow on that day, which reduces the spatiotemporal dispersion of the detection results. This is very important for rare species conservation and invasive species monitoring [20]. However, the advantage of eRNA timeliness is not universal. When the water temperature rises, the advantage of eRNA will fade, and the final degradation rate may be the same as eDNA (Figure 5). Temperature has been demonstrated as one of the key factors affecting the eDNA degradation rate. Exonuclease activity and microbial metabolism can be stimulated by a relatively high temperature environment (<50 °C), thereby influencing the eDNA degradation rate [29,30], while eRNA is much less affected by temperature. One possible explanation is that eRNA degrades more rapidly, overshadowing the effect of temperature. The eRNA degradation time only decreased from 22 h to 11 h when the temperature changed from 10 to 25 °C. It is generally believed that RNA is a far less stable molecule than DNA in vitro [20]. Since the timeliness of eRNA

is evaluated by the degradation rate of eDNA, it is necessary to analyze the dynamics of eDNA under different conditions.

Another crucial issue in eRNA degradation studies is whether the observations are generalizable. The researchers used different creatures to test eRNA degradation rates or degradation times in all the relevant studies, including this study, and arrived at varied conclusions, which may be due to the different target species. Attached organisms and swimming organisms, for example, have significantly distinct physiological structures and life behaviors. Swimming creatures will unavoidably collide during activities and their tissues will be more likely to fall off, increasing the concentration of eDNA/eRNA in water. Varied initial concentrations will result in different decay rates or tissue states shed by different species, which will alter the retention duration of eDNA/eRNA in water. The purpose of deploying eRNA in field measurement is to research the common eRNA rules among different species.

Furthermore, various factors influence eRNA in a complex natural environment. Although there is no decisive evidence regarding the particular environmental conditions that affect eRNA, we can draw inspiration from eDNA-related research and consider such factors as temperature, salinity, pH, UV intensity, dissolved organic carbon content, biological oxygen requirement, dissolved oxygen content, water flow, and so on [16,25,37,39]. This research has only examined the timeliness of eRNA as a temperature variable; other contributing factors need to be investigated.

Even if we demonstrate that eRNA has superiority over eDNA in spatiotemporal dispersion, it is not advisable to directly replace the monitoring of eDNA with eRNA. The timeliness of eRNA will also cause problems with false negatives due to the rapid degradation of eRNA. When researchers use eRNA to monitor organisms in a certain water body, they may conclude that the species has left or no longer populates the water body because the eRNA cannot be detected and this may not be so. Therefore, it is necessary to use eDNA for re-examination. Combining eDNA and eRNA and comprehensively analyzing the detection results could effectively avoid this problem.

## 5. Conclusions

In this paper, we have reported one of the first studies elucidating eDNA/eRNA degradation processes at different temperatures. The results showed that eRNA was less sensitive to temperature changes than eDNA. At low temperatures, the eRNA degradation rate was significantly faster than that of eDNA ($p = 0.016$ *), but with an increase in temperature the degradation rate constants of eRNA and eDNA tended to be more consistent and the differences between the two became smaller. It can be seen that eRNA has stronger timeliness than eDNA at low temperatures, and it can effectively reduce detection errors in eDNA analysis or result errors caused by an excessively long eDNA residual time. Furthermore, in order to avert the problem of the target species not being identified due to the rapid degradation of eRNA, it is necessary to use eDNA technology to re-examine eRNA detection results [39]. In conclusion, the incorporation of eRNA analysis into eDNA technology can not only increase the detection rate of genetic material from target species but can also enable better explanations of positive detection results [40]. Understanding the mechanisms that affect eDNA/eRNA degradation can improve the accuracy of molecular methods on a spatiotemporal scale as well as make the biological monitoring results more scientific and reliable; thus, this technique has the potential to become a powerful tool for stock assessment and fisheries conservation.

**Author Contributions:** Conceptualization, T.Q. and W.W.; methodology, T.Q.; software, T.Q.; validation, W.W.; formal analysis, T.Q.; resources, X.S. and X.J.; data curation, T.Q.; writing—original draft preparation, T.Q.; writing—review and editing, X.S. and W.W.; visualization, T.Q.; supervision, X.S. and X.J.; project administration, X.S.; funding acquisition, X.S. and X.J. All authors have read and agreed to the published version of the manuscript.

**Funding:** This research was funded by the National Natural Science Foundation of China [31872692]; Shandong Taishan Scholars Project and Innovation Team of Fishery Resources and Ecology in Yellow Sea and Bohai Sea, the Chinese Academy of Fishery Sciences [2021TD01] and the Central Public-Interest Scientific Institution Basal Research Fund, YSFRI, CAFS [No. 20603022021002].

**Institutional Review Board Statement:** The study was conducted according to the guidelines of the Declaration of Helsinki. All sampling protocols, as well as fish rearing practices, were reviewed and approved by Institutional Animal Care and Use Committee (IACUC) of the Yellow Sea Fisheries Research Institute (14 March 2019).

**Informed Consent Statement:** Not applicable.

**Data Availability Statement:** Not applicable.

**Acknowledgments:** The authors thank Su Chengcheng for sample collection and Sam Liu for data analysis.

**Conflicts of Interest:** The authors declare no conflict of interest.

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
