# Peer review of "Effects of Temperature on the Timeliness of eDNA/eRNA: A Case Study of Fenneropenaeus chinensis"

_water, doi:10.3390/w14071155_

Round 1

Reviewer 1 Report

This research is valuable. Under artificial experimental conditions, the effect of temperature was revealed. Since the natural seawater used for the experiments is purified, the eRNA may be relatively stable. It will be applied in the field environment in the future, so it is still necessary to discuss the possible variable influence in the discussion. Also, what is the number of replicates at each sampling time? Figures 5 and 6, and Tables 2 and 3, do not have statistical values (eg SD?), which may greatly affect the reliability of this study.

Line 117 mm? g?

Line 128 purified natural seawater, how to purify? with 0.3um filter?

Author Response

We gratefully appreciate for your valuable suggestion.

You mentioned that it is necessary to discuss the possible variable influence in the discussion, I quite agree with your suggestion. Like eDNA, eRNA is affected by a variety of factors in the field environment, causing changes in its persistence time. Regarding eDNA, studies have confirmed the response mechanism of eDNA under the influence of temperature, UV-B, and pH. However, until our work was carried out, there are few related articles on the factors that affect the degradation rate of eRNA. As far as I know, we are one of the first research group to carry out this work. Perhaps UV-B and pH also affect the degradation of eRNA, and may also be affected by other factors. This requires more research as support, and is also the core of our next work. We have added this part to the discussion of manuscript.  

We performed three replicate experiments, and each water sample also performed three qPCR replicates to ensure the reliability of the results, which may be misunderstood due to the unclear expression in the manuscript, we explained more detail in lines 163-166 of the manuscript, and updated information in the flow diagram (Figure 1). SD values are added in Tables 2 and 3 to prove the reliability of our data. Figures 5 and 6 are fitted by bringing the mean of the original data into the model. R2 can explain the fitting effect.  

Regarding the issue of the line 117, we have added unit of weight and length in the manuscript, this is a negligence of our work, thank you for pointing it out.

Regarding the treatment of purified natural seawater. We transported the natural seawater back to the laboratory, put it into the reservoir, exposed to sunlight, and then filtered through a 0.45 µm glass fiber filters. Before the experiment, the reservoir was soaked in 5% potassium permanganate solution, and thoroughly rinsed with clean water. The seawater is always aerated to ensure dissolved oxygen content during storage.

Thank you again for your positive and constructive comments and suggestions on our manuscript! Finally, I wish you all the best in your work.

Reviewer 2 Report

Qian and colleagues have present their work entitled "Effects of temperature on the timeliness of eDNA/eRNA: A case study of Fenneropenaeus chinensis." This study can be interesting and important that could fit the scope of the journal. However, some major issues need to be addressed before considering publishing their work. 

Q1. Is the experimental design appropriate for the studies of eRNA? In your case, eRNA has been dramatically degraded within 4 hours, and the concentration of eRNA is subsequently constant than that of eDNA in response to temperature changes. The author concluded that eRNA showed better under temperature change and maintained a faster degradation rate at low temperatures, which seems to be questionable. This conclusion is only valid for the concentration of eRNA after 4 hours in this work. The concentration of eRNA under different temperature treatments were not determined within 4 hours, which is very likely that the concentration of eRNA will be sensitive to temperature changes over the fine-scaled time course within 4 hours

Q2. The concentration of eDNA and eRNA in the tank with different temperature treatments was estimated. Did you perform replicate sampling? In table 2, you mentioned the average concentration of eDNA and eRNA. The standard deviation shall be provided. What is the sample size?

Q3. What is the difference between Figure 5 and Figure 6 ? There is no need to show two different figures with the same information.

Q4. The novelty of this work is the effects of temperature on eDNA/eRNA specific for the species you examined. In fact, the effects of environmental factors on eDNA/eRNA have been extensively investigated in many studies. The author shall compare the results with other studies. For example, the degradation rate of eDNA/eRNA in response to temperature changes is species-specific ?

Q5. How do you expect results from this can be applied to the eDNA/eRNA studies in the field? In the wild, eDNA and eRNA will not be affected by the temperature alone.

Author Response

We gratefully appreciate for your valuable suggestion.

Answer 1

I believe that our experimental design is suitable for eRNA, because there is precedent for applying this method to eRNA research, and we referenced and improved their research methods for experiments. The relevant literatures are as follows:

Wood, S.A., Biessy, L., Latchford, J.L., Zaiko, A., von Ammon, U., Audrezet, F., Cristescu, M.E., Pochon, X., 2020. Release and degradation of environmental DNA and RNA in a marine system. Sci. Total Environ. 704, 135314. https://doi.org/10.1016/j.scitotenv.2019.135314.

Marshall, N.T., Vanderploeg, H.A., Chaganti, S.R., 2021. Environmental (e)RNA advances the reliability of eDNA by predicting its age. Sci. Rep. 11, 2769. https://doi.org/10.1038/s41598-021-82205-4.

       This is a good question to ponder that the mentioned eRNA may be temperature sensitive within 4 hours. There is a possibility that the temperature sensitivity of eRNA is masked by rapid degradation of eRNA, and eRNA may be sensitive to temperature changes within 4 hours, which is a problem that needs to be investigated. But it has to be mentioned that our conclusion is for a large-scale time course, and the effect of eRNA concentration changes within 4 hours on stock investigation and biomass assessment is not as large as expected. During large scale offshore operations, the concentration changes of fine-scale time are easily missed, so we did not set sampling within 4 hours, but set sampling time points within 24 hours as much as possible to avoid missing the eRNA degradation process. Nonetheless, your comments will be of great help to our follow-up studies, and we believe that the changes in eRNA concentration over 4 hours are valuable information. In the future, we will carry out targeted work to study the changes in eRNA concentration over fine-scale time course.

Answer 2

We performed three replicate experiments, each water sample also performed three qPCR replicates to ensure the reliability of the results, possibly due to misunderstanding caused by the unclear expression of the article, we have added a more detailed description in lines 163-166 of the manuscript, and relevant parts has been added to the flowchart (Figure 1). SD values are added in Tables 2 and 3 to demonstrate the credibility of our data. Total 129 samples in this manuscript, including 120 test samples and 9 negative controls.

Answer 3

Figure 5 is the comparison between eRNA and eDNA at the same temperature, and Figure 6 is the change of eRNA or eDNA at different temperatures. After our deliberation, we finally decided to add Figure 6, hoping that the response of eDNA and eRNA to temperature can be more intuitively represented by Figure 6.

Answer 4

This is a good opinion. Currently, eRNA-related studies, including this manuscript, have chosen different target species, including ascidians, shellfish, crustaceans, and fish. It has to be admitted that the conclusions of this manuscript may be species-specific, but our purpose is to draw general conclusions while studying a specific species of organisms, which is the ultimate goal of most scientific research, but before that, compare the results with other studies are essential. In previous manuscript we compared the results with those in the studies of Wood et al (2019) and Marshall et al (2021). Now, we have added a species-specific correlated discussion on lines 404-414.

Answer 5

I agree with your opinion very much. This study chose the more intuitive temperature as the research variable, but in a complex natural environment, eRNA is not only affected by temperature, and there are many other environmental factors. There is no research on what the environmental factors that affect eRNA are, but there are many related studies on eDNA, we can draw inspiration from them, such as salinity, pH, UV intensity, dissolved organic carbon content, biological oxygen requirement, dissolved oxygen content, water flow and so on, which require more research to explore, we have added this discussion to lines 415-421 of the manuscript.

Thank you again for your time and suggestions on this manuscript!

Kind Regards

Reviewer 3 Report

Literature is up to date. The reader can easily understand how this manuscript is important to fill the gap in the literature. You can also add additional references to improve the introduction and discussion. 

1)Kaede Miyata, Yasuaki Inoue, Yuto Amano, Tohru Nishioka, Masayuki Yamane, Takamitsu Kawaguchi, Osamu Morita, Hiroshi Honda,
Fish environmental RNA enables precise ecological surveys with high positive predictivity, Ecological Indicators,Volume 128,2021,
https://doi.org/10.1016/j.ecolind.2021.107796.
2) Heather D. Veilleux, Melissa D. Misutka, Chris N. Glover,
Environmental DNA and environmental RNA: Current and prospective applications for biological monitoring,Science of The Total Environment,
Volume 782,2021,146891,
https://doi.org/10.1016/j.scitotenv.2021.146891.

other comments: (see the pdf file)

Line 87: Chinese white shrimp (write the English name).

Line 116: it may be better to make a flow diagram that shows the steps. (Like Figure 1)

Line 284: reference is only for eDNA. 

Lines 289-290: better to seperate references for eDNA(barnes, dejean,pilliod, strickler, thomsen)  and eRNA (marshall, wood)

Lines 324-331: this finding is also important and write it again with one sentence in the conclusion part. 

Author Response

Thank you very much for your valuable comments and suggestions.

The two references you mentioned are extremely useful for our research. These studies were not included in the references because they had not yet been published at the time this article was written. We updated the discussion part and added them to the reference part. Thank you very much for your advice.

Relevant information about the name of the research species has been added to the line 100 of the article. To avoid confusion with other prawns, the target organism is still referred to by its Latin name in the article.

We added new information to the original flowchart (Figure 1), which more clearly illustrates the experimental process, while maintaining the fluency of reading.

Added eRNA related reference to line 284 (Pochon et al., 2017)

Pochon, X., Zaiko, A., Fletcher, L.M., Laroche, O., Wood, S.A., 2017. Wanted dead or alive? Using metabarcoding of environmental DNA and RNA to distinguish living assemblages for biosecurity applications. PLOS ONE 12, e0187636. https://doi.org/10.1371/journal.pone.0187636.

We adopted your suggestion to separate eDNA and eRNA related references in lines 289-290. In future work, we will pay attention to this kind of problem.

A description similar to lines 324-331 has been added to the conclusion part.

Thank you again for your time and suggestions on this manuscript!

Kind Regards

Round 2

Reviewer 1 Report

It is ok now.

Author Response

Dear reviewer,

Thank you  for your positive and constructive comments and suggestions on our manuscript! Finally, I wish you all the best in your work.

Reviewer 2 Report

The authors have replied to my previous comments and revised the manuscript accordingly. However, some issues are still required to address as follows:

In table 2, the eDNA concentration with the temperature treatments of 20 and 25 0°C dramatically decreased after 72h. However, their eDNA concentration after 336 and 504 h were shown to be higher than those after 72 and 120 h. Any possible explanation for these results? 

Additionally, the proofreading of this manuscript is also required to address some issues as follows:

Line 96: it isurgent ? 

Line: 376-379: These two citations look the same for me. What is the difference between Ficetola et al., 2008a and 2008b?

Line 432-435: These two citations look the same for me. What is the difference between Thomsen et al., 2012a and 2012b?

Line 436-436: The title of this citation: Monitoring endangered freshwater biodiversity using environmental DNA: SPECIES MONITORING BY ENVIRONMENTAL DNA: Is it correct?

Author Response

       Thank you very much for your valuable comments and suggestions.

       Regarding your query about the abnormally high eDNA concentration. The content of eDNA in water is lower due to faster degradation of eDNA at high temperatures. Therefore, the number of cycles required for qPCR is more. The Ct value of eDNA has been greater than 30 after 12 hours, according to the Ct value returned by our equipment. Small errors are magnified, and Ct values are less repeatable as a result . Combining the above factors, we defined 10 copies/mL as the detection limit, and below 10 copies/mL was considered complete degradation. We do not think this phenomenon needs to be discussed in the manuscript, thank you for your question.

We have revised the original manuscript without noticing that the space on line 96 has also been removed, which has been revised in the manuscript and checked elsewhere to prevent the same problem.

       It may be due to the reference management software we used that duplicated/misidentified individual references after revisions in the original manuscript, we have corrected the errors in lines 376-379 and 432-435 and checked other references.

       We have removed SPECIES MONITORING BY ENVIRONMENTAL DNA from the reference on line 436 to ensure proper citation.

       Thank you again for your time and careful review on this manuscript!

       Kind Regards